# Increased Pathway Complexity Is a Prognostic Biomarker in Metastatic Castration-Resistant Prostate Cancer

**DOI:** 10.3390/cancers13071588

**Published:** 2021-03-30

**Authors:** Bram De Laere, Alessio Crippa, Ashkan Mortezavi, Christophe Ghysel, Prabhakar Rajan, Martin Eklund, Alexander Wyatt, Luc Dirix, Piet Ost, Henrik Grönberg, Johan Lindberg

**Affiliations:** 1Department of Human Structure and Repair, Ghent University, 9000 Ghent, Belgium; piet.ost@ugent.be; 2Cancer Research Institute Gent (CRIG), Ghent University, 9000 Ghent, Belgium; 3Department of Medical Epidemiology and Biostatistics, Karolinska Institutet, 17177 Stockholm, Sweden; alessio.crippa@ki.se (A.C.); ashkan.mortezavi@ki.se (A.M.); martin.eklund@ki.se (M.E.); henrik.gronberg@ki.se (H.G.); johan.lindberg@ki.se (J.L.); 4Department of Urology, University Hospital Basel, 4031 Basel, Switzerland; 5Department of Urology, AZ Sint-Jan AV Brugge-Oostende, 8000 Brugge, Belgium; christophe.ghysel@azsintjan.be; 6Centre for Molecular Oncology, Barts Cancer Institute, Queen Mary University of London, London E1 4NS, UK; p.rajan@qmul.ac.uk; 7Department of Urologic Sciences, Vancouver Prostate Centre, University of British Columbia, Vancouver, BC V6H 3Z6, Canada; awyatt@prostatecentre.com; 8Department of Oncology, GZA Hospitals Sint-Augustinus, 2610 Antwerp, Belgium; luc.dirix@telenet.be; 9Center for Oncological Research (CORE), University of Antwerp, 2610 Antwerp, Belgium; 10Department of Radiation Oncology, GZA Hospitals Sint-Augustinus, 2610 Antwerp, Belgium; 11Science for Life Laboratory, 17177 Stockholm, Sweden

**Keywords:** mCRPC, biomarker, cfDNA

## Abstract

**Simple Summary:**

Circulating tumour DNA profiling can cost-efficiently accelerate biomarker discovery within oncology trials. However, biomarker identification in metastatic castration-resistant prostate cancer is confounded by a heterogeneous genomic landscape with few commonly-perturbed genes and a large number of infrequently mutated, yet potentially biologically-relevant, cancer drivers. Hence, large sample sizes are required for the stratified evaluation of these infrequent perturbations. To circumvent this issue, we investigated whether grouping genomic alterations with other events within the same cellular pathways would offer increased precision for biomarker discovery. We undertook an individual patient-level pooled analysis of 342 patients with metastatic castration-resistant prostate cancer-initiating abiraterone acetate or enzalutamide. We found that the total number of altered pathways, which we termed the pathway complexity index (PCI) was associated with a poor prognosis. Since genomic profiling is now standard practice in interventional oncology trials, our findings highlight the importance of comprehensive genomic profiling for biomarker discovery and utilization.

**Abstract:**

Metastatic castration-resistant prostate cancer (mCRPC) is a heterogeneous disease, characterized by common and rare driver gene alterations that provide a selective growth advantage for progressing tumour cells. We hypothesized that the number of distinct gene driver alteration-affected pathways or gene classes was associated with poor prognosis in patients initiating androgen receptor signalling inhibitors (ARSi). We performed a post hoc analysis of an amalgamated baseline circulating tumour DNA (ctDNA) mutational landscape dataset of ARSi-treated men with mCRPC (*n* = 342). We associated the detected hotspot, pathogenic, and/or high impact protein function-affecting perturbations in 39 genes into 13 pathways. Progression-free (PFS) and overall survival (OS) were analysed using Kaplan–Meier curves and multivariate Cox regression models. Driver gene alterations were detected in 192/342 (56.1%) evaluable patients. An increased number of affected pathways, coined pathway complexity index (PCI), resulted in a decremental PFS and OS, and was independently associated with prognosis once ≥3 pathway or gene classes were affected (PFS HR (95%CI): 1.7 (1.02–2.84), *p* = 0.04, and OS HR (95%CI): 2.5 (1.06–5.71), *p* = 0.04). Additionally, visceral disease and baseline PSA and plasma ctDNA levels were independently associated with poor prognosis. Elevated PCI is associated with poor ARSi outcome and supports comprehensive genomic profiling to better infer mCRPC prognosis.

## 1. Introduction

Comprehensive genomic profiling of metastatic castration-resistant prostate cancer (mCRPC) has demonstrated extensive inter-patient heterogeneity, with the identification of relatively few commonly perturbed genes and a large number of infrequently mutated cancer drivers in the so-called “long tail” [1]. Consequently, any precision medicine feasible trial will be underpowered for the stratified evaluation of the majority of reported driver genes. The mCRPC genomic landscape in metastatic tissue is faithfully mirrored in plasma-derived circulating tumour DNA (ctDNA) [2] or circulating tumour cells [3]. Our prior work demonstrated that ctDNA profiling can identify microsatellite instability (MSI), genomic structural rearrangements (GSRs), and that *TP53* alterations can stratify mCRPC patients on androgen receptor signalling inhibitors (ARSi) into clinically relevant prognostic groups [4,5]. Thus, the potential clinical utility of minimally-invasive liquid biopsies to identify driver DNA alterations has been demonstrated and has set the stage for precision urologic oncology trials, aiming towards improving patient prognostication and clinical management.

A prerequisite for identifying an association between *TP53* perturbations and ARSi prognosis was the high (24.8%) frequency of *TP53*-altered patients [5]. Although *TP53* is the most frequently mutated gene in cancer, alterations in *TP53* have the same basic consequence as uncommonly altered oncogenic genes; namely to provide a selective growth advantage. Most cancers harbour alterations in a handful of driver genes, but heterogeneity exists. The known drivers of cancer, commonly or uncommonly altered, are all associated with a limited number of cellular signalling pathways (e.g., AR pathway, PI3K pathway, cell cycle signalling, DNA damage repair, etc.) which in turn affect core cellular processes, resulting in increased cancer cell fitness [6].

Although the sizes of currently existing datasets prevent clinical evaluation of genes in the long tail, we hypothesized that an increase in the number of perturbed distinct pathways in an individual cancer, contributed to by common and uncommon driver genes, will lead to an increase in biological versatility. Therefore, we investigated if the number of altered pathways, coined the pathway complexity index (PCI), was associated with prognosis in men with mCRPC starting ARSi.

## 2. Results

### 2.1. Study Patient Populations

A baseline ctDNA mutational landscape dataset from abiraterone- or enzalutamide-treated men with MSI- and/or hypermutator-negative mCRPC (*n* = 342), recruited between 2014 and 2017, was assembled using publicly available datasets (i.e., our CORE/PROBIO cohort study (*n* = 142) [4,5] and the prospective NCT02125357 trial (*n* = 200) [7]) (Figure 1). Whilst the Vancouver Prostate Centre (defined as VANC) encompassed 200 mCRPC patients starting ARSi as first-line therapy, the CORE/PROBIO-enrolled patients also consisted of 61/142 (43%) and 19/142 (13.4%) patients previously treated with chemotherapy and ARSi, respectively (Appendix A). We observed a higher median ctDNA fraction (9.15% vs. 3.90%, *p* < 0.001) in the CORE/PROBIO cohort compared with the VANC cohort. However, when comparing clinical features within the subgroup of treatment-naive patients (*n* = 277) from both cohorts, no differences were observed (Appendix A). Overall, we did not observe significant differences between progression-free (PFS) (median, 6.67 vs. 7.40 months, *p* = 0.23) and overall survival (OS) (median, 24.8 vs. 20.3 months, *p* = 0.09) estimates between the CORE/PROBIO and VANC cohorts respectively (Appendix A). Nonetheless, due to the observed variability between both cohorts in terms of prior systemic therapy exposure and ctDNA fractions, cox-regression models are stratified by cohort to allow for differences in cohort-specific hazards.

### 2.2. The ctDNA Landscape of Driver and Pathway Alterations in mCRPC

Driver gene perturbations were detected in 192/342 (56.1%) evaluable patients at baseline (Appendix A). Overall, *AR* (135/342, 39.5%), *TP53* (93/342, 27.2%), *PTEN* (43/342, 12.6%), *FOXA1* (20/342, 5.8%), and *SPOP* (20/342, 5.8%) were the most commonly perturbed genes (Figure 2A, Appendix A). For these top 5 perturbed genes, the prevalence in both cohorts was similar except for *PTEN* perturbations, which were higher in the CORE/PROBIO cohort in comparison to the VANC cohort (19.7% vs. 7.5%, respectively, *p* = 0.001) (Appendix A). When these genes were grouped into their defined pathways and/or gene classes, the *AR* pathway (139/342, 40.6%), *TP53* gene class (93/342, 27.2%), the PI3K pathway (54/342, 15.8%), the cell cycle signalling (37/342, 10.8%), and DNA repair (32/342, 9.4%) were the most commonly perturbed pathways or gene classes (Figure 2B, Appendix A). No driver alterations were detected in *MSH2*, *CDK4*, *NFE2L2*, *IDH1*, *FANCG*, *GNAS*, *FBXW7*, and *IDH2*.

### 2.3. Individual Pathways or Gene Classes and Outcome

In univariate analysis, the median PFS and OS estimates were significantly shortened in patients carrying *AR* pathway, *TP53*, PI3K pathway, cell cycle signalling, and DNA repair driver perturbations (all *p* < 0.01) (Appendix A). For OS, we additionally observed WNT/*β*-catenin pathway and transcription factor perturbations to be associated with inferior prognosis (all *p* < 0.05) (Appendix A). The majority of these observations, except for DNA repair, were recapitulated in both cohorts (Appendix A). Multivariate (MV) analysis of an individual pathway or gene class, corrected for PSA and ctDNA levels, prior chemotherapy and ARSi exposure, and presence of visceral metastases, revealed that the *TP53* and DNA repair gene classes maintained independent prognostic value for both PFS and OS (Appendix A). This was preserved when the MV analysis for PFS and OS incorporated all pathway or gene classes (Appendix A). In both models, prior ARSi exposure, presence of visceral metastases and elevated ctDNA and PSA levels were also associated with inferior prognosis. Interestingly, when putting the *TP53* and DNA repair gene classes head to head, and allowing for interaction with the underlying cohorts, we observed that the PFS prognostic value of the *TP53* gene class could be recapitulated in both cohorts, whereas the association of the DNA repair category with PFS and OS was only observed in the VANC cohort (Appendix A).

### 2.4. Pathway Complexity Index (PCI) and Outcome

When associating the number of the pathway or gene class hits with outcomes we observed a decremental gradient in PFS and OS as the number of affected pathways or gene classes accumulated (Appendix A). We observed how outcome estimates did not differ between patients with 1–2 and ≥3 affected pathways or gene classes. Grouping these patients resulted in 123/342 (36%) and 69/342 20.2%) patients with a pathway complexity index (PCI) of 1–2 and ≥3, respectively, with decremental PFS (median, 12.5 vs. 5.8 vs. 3.2 months, *p* < 0.0001) and OS (median, not reached vs. 18.2 vs. 9.7 months, *p* < 0.0001) survival curves in comparison to patients without driver pathway hits (Figure 3A,C).

No frequency differences were observed between both cohorts (Chi-square: *p* = 0.10) (Appendix A). In multivariate analysis the PCI, using patients without driver pathway hits (PCI = 0) as a reference, carried independent PFS (HR 1.7, 95%CI 1.02–2.84, *p* = 0.04) (Figure 3B) and OS (HR 2.5, 95%CI 1.06–5.71, *p* = 0.04) (Figure 3D) prognostic value once ≥3 pathway or gene classes were affected. Additionally, we observed how, besides a high PCI the presence of visceral metastases, serum PSA and plasma ctDNA levels (all *p* < 0.001) were independently associated with inferior outcomes. Importantly, incorporating and modelling the PSA and ctDNA fraction as continuous variables within the Cox regression model (Figure 3C,D, right) revealed how increasing levels of serum PSA or plasma ctDNA resulted in a strong positive correlation with the time-to-event hazard ratio’s, i.e., the higher the PSA or ctDNA level at baseline, the higher the chance of an event (progression or death) occurring.

The decremental PFS and OS with increasing PCI were preserved in the prior treatment-naive cohort-stratified subgroup analysis (Appendix A). In an exploratory analysis, we looked into recurrent combinations in poor prognosis patients with ≥3 affected pathways or gene classes (*n* = 69), which revealed that 44/69 (63.8%) patients harboured combined *AR* pathway and *TP53* gene class perturbations, of whom 24/44 (54.5%) carried an additional perturbation within the PI3K pathway (Appendix A). Surprisingly, alterations in genes of the same pathway (thought to be evolutionary redundant [6]), were not mutually exclusive in all patients. This warrants future investigation of the potential prognostic value of the number of altered driver genes that overlap in the same signalling pathway (Appendix A).

## 3. Discussion

We demonstrate how plasma ctDNA profiling can identify subpopulations of patients with mCRPC that have accumulated multiple hits in cancer driver pathways, which is independently associated with poor ARSi outcome and survival. Our study has some limitations. Firstly, this retrospective analysis is hypothesis-generating and was not designed to assess the effect of (sub)clonal mutations; which likely harbour prognostic information in the context of the disease trajectory and therapeutic selection pressure [8], nor to benchmark the prognostic value of PCI relative to tumour mutational burden (TMB). The relatively small sizes of our panels cannot provide a robust TMB estimate [9], unless in the context of microsatellite instability or hypermutation [4,7]. Additionally, the CORE/PROBIO study had incomplete data collection on clinical features and blood-based parameters (e.g., Gleason score, number of metastatic lesions, comorbidities, ECOG, LDH, etc.), which are associated with prognosis [10]. Furthermore, the merged variant dataset was generated from overlapping regions of two hybridisation-based capture assays with different capture designs, which could be exemplified by the rate of *PTEN* perturbations. This was markedly higher in the CORE/PROBIO cohort as a result of the intronic sequencing performed for the CORE/PROBIO patients, enabling the detection of GSRs [11]. Finally, these data were derived from different patient populations. Although similar outcome estimates were observed, we took the inherent differences into account in our cohort-stratified time-to-event analyses. Higher ctDNA fractions were detected in the CORE/PROBIO cohort in comparison to the treatment-naive Vancouver (VANC) patients, which can be in part explained by the all-comer nature of the former cohort, where a group of patients initiated ARSi at a later line of therapy. In the context of genomic biomarkers, this underlying demographic difference was most clearly demonstrated for the association between the DNA repair deficiency status and ARSi outcome. Although a significant association in multivariate analysis was observed in the total population, this was driven by the Vancouver patients. Genes with different roles in the DNA repair process (e.g., sensing, signalling or repair) can cause distinct mutational signatures [12] and responses to, e.g., PARP inhibition [13]. It is, therefore, likely that the interplay with endocrine therapy may differ between individual genes. The CORE/PROBIO-cohort has a significantly lower proportion of men with pathogenic *BRCA2* germline alterations and instead demonstrates increased levels in *ATM* and *CHEK2* [4]. The differences in clinically relevant germline DNA repair alterations may have contributed to the differences observed here and warrants further investigation in future biomarker-driven studies. Other tissue biopsy-based studies have suggested *SPOP* mutations [14] and biallelic inactivation of *RB1* [15] to be independently associated with good and poor outcomes of ARSi, respectively. Our ctDNA data revealed no association for SPOP and demonstrated how perturbed cell cycle signalling, in general, was associated with poor ARSi outcomes in univariate, but not in multivariate, analysis. We did not study *RB1* alone due to its low prevalence (i.e., 10/342 (2.9%) cases). Importantly, *RB1* intronic sequencing was not carried out in the plasma samples from the Vancouver cohort, which may have led to a misclassification of a subset of patients as not carrying a driver *RB1* event, due to an inability of detecting inactivating GSRs. Additionally, it is unclear if Abida et al. [15] incorporated correction for tumour burden estimates in the multivariate analysis, which questions the current generalizability of RB1′s independent association with ARSi outcomes, especially since successful biopsy profiling is correlated with high tumour burden [16]. Although a different OS definition and patient population were used in comparison to Chen et al. [17], we were able to recapitulate the association between *RB1* bi-allelic inactivation and poor overall survival using the CORE/PROBIO-cohort data (Appendix A). To the best of our knowledge, these data represent the first demonstration of the clinical validity of the pathway complexity index, which dictates the prognosis of mCRPC patients on an upcoming ARSi therapy. In patients with a high PCI, we observed how *AR* and *TP53* perturbations most commonly co-occurred, with >50% of those patients having a PI3K pathway perturbation as well. Targeting both the *AR* and PI3K pathway was tested in a randomised controlled trial, and demonstrated durable responses in *PTEN*-null tumours when treated with abiraterone and the Akt-inhibitor ipatasertib [18]. The reported data warrants the continued investigation of new combination therapy modalities or biomarker signature-therapy matches in large liquid biopsy-driven multi-arm prospective clinical studies, such as PC-BETS (ClinicalTrials.gov Identifier: NCT03385655) [19] and the randomized clinical trial ProBio (ClinicalTrials.gov Identifier: NCT03903835) [20], where longitudinal monitoring will make it possible to study the PCI dynamics throughout the course of multiple therapeutic interventions. Additionally, with the decreasing cost of DNA sequencing and increasing DNA capture panel sizes we anticipate that the novel PCI metric will be subject to continued evaluation and fine-tuning in future biomarker-driven research.

## 4. Materials and Methods

### 4.1. Patient Cohort and ctDNA Dataset Generation

A baseline ctDNA mutational landscape dataset from abiraterone- or enzalutamide-treated men was assembled using publicly available datasets (i.e., our CORE/PROBIO cohort study (*n* = 142) [4,5] and the prospective ClinicalTrials.gov Identifier: NCT02125357 trial (*n* = 200) [7]) (Figure 1). All reported genomic alterations, encompassing copy-number aberrations, mutations and GSRs, within 39 genes that overlapped between both datasets were merged. The following filters were applied to the annotated variants to generate a list of driver genomic alterations: (1) For indel and missense variants: only known hotspots, pathogenic, and/or annotated as high-impact variants by the Variant Effect Predictor (PMID: 27268795) were kept; (2) Deleterious copy-number aberrations need to encompass deep and/or homozygous deletions (log ratio <–1.0); (3) Only high-level copy-number amplifications (log ratio > 0.6) were retained, except for *AR*, where any gain was annotated as a driver hit in the *AR* gene; and (4) Only significant genomic structural rearrangements (GSRs), resulting in loss of- or gain of protein function were kept.

The driver genomic alterations were grouped into pathways except for *TP53*, *SPOP*, *IDH1*, and *CDK12* who were kept as individual gene classes. *TP53* is involved in multiple pathways and hard to assign to a specific pathway [21], *SPOP* and *IDH1* represent distinct prostate cancer subtypes [22], as does *CDK12* [23]. The pathway and driver complexity index were defined as the number of detectable driver genomic alterations in the selected 39 genes, which in turn were associated with 13 pathways or gene classes: AR pathway (*AR*, *FOXA1*, *ZBTB16*), TP53 class (*TP53*), PI3K pathway (*AKT1*, *PIK3CA*, *PIK3CB*, *PIK3R1*, *PTEN*), cell cycle signalling (*RB1*, *CCND1*, *CDKN1B*, *CDKN2A*, *CDK4*, *MYC*), DNA repair deficiency (*BRCA1*, *BRCA2*, *ATM*, *FANCA*, *MLH1*, *MSH2*, *MSH6*), WNT/b-catenin pathway (*APC*, *CTNNB1*), SPOP class (*SPOP*), chromatin modifiers (*CHD1*, *KDM6A*, *ARID1A*), CDK12 class (*CDK12*), transcription factors (*MED12*, *NFE2L2*, *NKX3.1*), MAPK pathway (*KRAS*, *BRAF*), IDH1 subtype (*IDH1*), and other (*FBXW7*, *GNAS*, *IDH2*, *FANCG*). The ctDNA fraction was estimated as described previously [4,7].

### 4.2. Statistical Analysis

The current post hoc analysis evaluated progression-free (PFS) and overall (OS) survival as clinical endpoints. PFS was defined as time to first confirmed PSA progression, clinical and/or radiologic progression, or death from disease. Overall survival (OS) outcome was defined as the time from baseline blood sampling to death from any cause. Survival curves for the individual and the number of affected pathways or gene classes (i.e., PCI) were estimated by Kaplan–Meier (KM) analysis. Survival differences were determined using the log-rank test. The effects of pathway/gene class alterations and PCI were quantified by hazard ratios (HRs) in cohort-stratified multivariate Cox regression models, adjusted for the following covariates: baseline PSA and ctDNA levels, prior chemotherapy, prior ARSi exposure, and presence of visceral metastases. The baseline ctDNA fraction and PSA concentration were incorporated in the multivariate model as continuous variables, and were modelled in the Cox regression model using restricted cubic splines. The association of baseline ctDNA fraction and PSA concentration as continuous variables with the HRs for time-to-event outcomes (PFS and OS) were graphically presented using their median values as reference points. Differences between the two cohorts were assessed using the Mann–Whitney U (continuous variable) and Chi-square tests (categorical variables). All tests were performed in R (v.1.1.463), with a two-sided *p*-value < 0.05 as being considered as statistically significant.

## 5. Conclusions

We demonstrate for the first time that the elevated pathway mutational complexity is independently associated with poor prognosis, which warrants comprehensive genomic profiling for the prognostication of patients with advanced prostate cancer.

## Figures and Tables

**Figure 1 cancers-13-01588-f001:**
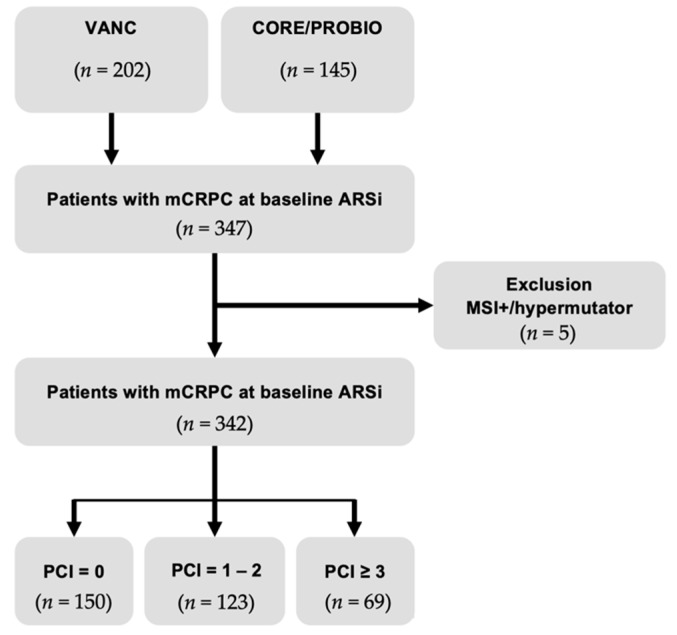
CONSORT diagram of patients with metastatic castration-resistant prostate cancer (mCRPC) treated with androgen receptor signalling inhibitors (ARSi). A total of 342 patients with mCRPC who underwent cell-free DNA profiling received ARSi (i.e., abiraterone acetate or enzalutamide). Five patients were excluded from downstream analysis as they were positive for the microsatellite instability (MSI) or hypermutator genotype. VANC, Vancouver Prostate Centre cohort; CORE/PROBIO, Centre for Oncological Research and Prostate Biomarker cohort; PCI, pathway complexity index.

**Figure 2 cancers-13-01588-f002:**
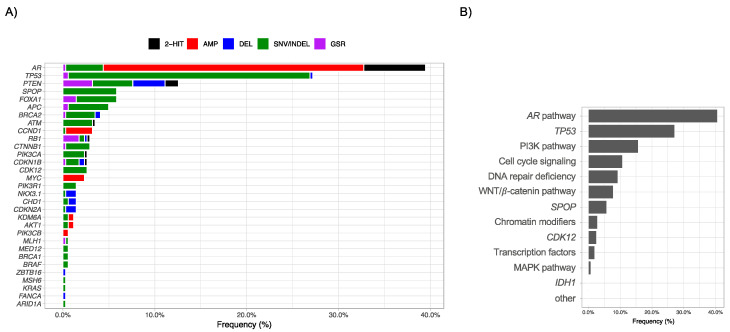
Prevalence of genomic alterations in baseline plasma ctDNA samples from patients with metastatic castration-resistant prostate cancer-initiating abiraterone or enzalutamide (*n* = 342). (**A**) Gene-level alteration frequencies, colour-filled according to perturbation type. (**B**) Pathway or gene class-level alteration frequency. Abbreviations: AMP, amplification; DEL, deletion; SNV/INDEL, single nucleotide variation, insertion or deletion; GSR, genomic structural rearrangement.

**Figure 3 cancers-13-01588-f003:**
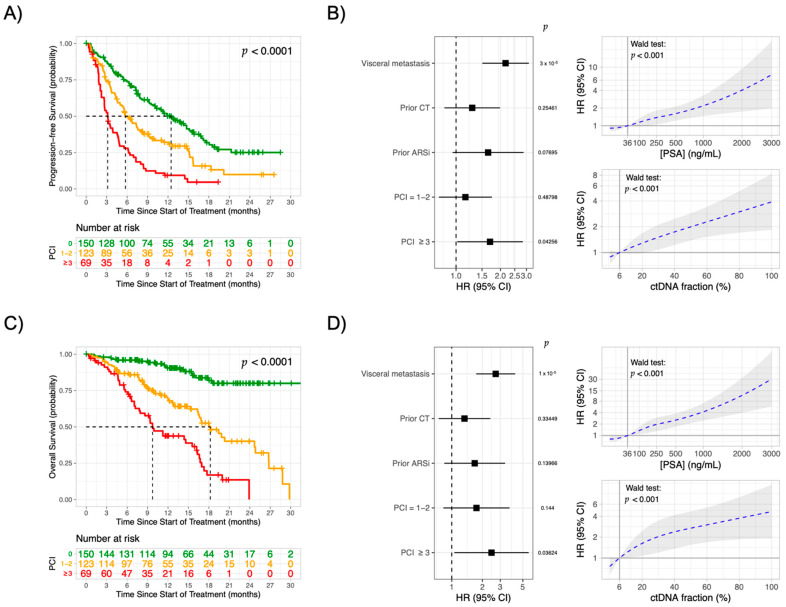
Increasing pathway complexity is associated with poor outcomes to abiraterone or enzalutamide in metastatic castration-resistant prostate cancer. Kaplan–Meier analysis of progression-free (**A**) and overall survival (**C**), stratified according to the pathway complexity index (PCI) at baseline. p-value is calculated via the log-rank test. Multivariate Cox regression analysis (hazard ratio (confidence interval)) of progression-free (**B**) and overall (**D**) survival using baseline characteristics and pathway complexity index (using a PCI = 0 as reference). Within the multivariate cox regression model, serum PSA and plasma ctDNA levels were incorporated and modelled as continuous variables using restricted cubic spline models. The correlation between increasing levels of serum PSA or plasma ctDNA and time-to-event hazard ratio’s (for progression-free (PFS) and overall survival (OS)) is graphically presented in Panel B and D (right), using the median PSA (36 ng/mL) and ctDNA fraction (6%) as reference points (i.e., HR = 1). The observed positive correlation was tested for significance using the Wald test (both *p* < 0.001). Abbreviations: ARSi, androgen receptor signalling inhibitors; CT, chemotherapy; ctDNA, circulating tumour DNA fraction; PSA, prostate-specific antigen. *p*-values calculated via Wald test.

## Data Availability

All data relevant for the interpretation of our findings is provided in the current and main manuscripts (i.e., De Laere et al. (PMID: 30209161) and Annala et al. (PMID: 29367197); or the Appendix A and methods for the manuscript, except for the raw sequence data. Any data providing genotype information is considered to be a personal registry by the Swedish law (Personal Data Act), thereby prohibiting the submission to a public repository. The raw sequence data is instead available upon request from the authors (For Belgium/Sweden: contact johan.lindberg@ki.se; For Canada: contact awyatt@prostatecentre.com) if approval has been obtained from a Regional Ethical Vetting Board.

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
