# Peer review of "Increased Pathway Complexity Is a Prognostic Biomarker in Metastatic Castration-Resistant Prostate Cancer"

_cancers, 2021, doi:10.3390/cancers13071588_

Round 1
Reviewer 1 Report
This paper evaluated the mutation landscape in mCRPC, treated with AR signaling inhibitors, pointed out the important rule of pathway complexity in PFS and OS. The association with alteration in P53, AR and PI3K pathways and akt inhibors durable responses is interesting. But all these results are not so novel and are already been described.
Reviewer 2 Report
In their manuscript „Increased pathway complexity is a prognostic biomarker in metastatic castration-resistant prostate cancer“ De Laere et al. make the case for the association of the number of observable genetic events in a ctDNA analysis with PFS and OS of prostate cancer patients undergoing ARSi treatment. This work quantifies the genetic events known to be associated with prostate cancer and calculates their impact on disease prognosis and adds to their previous observations of TP53 as a predictive genetic event in this setting. Albeit not completely novel, the authors elaborate their findings in two independent prostate cancer cohorts and show that the number of genetic aberrations beyond TP53 might influence disease prognosis. The manuscript is well written, the data and results support largely the statements made by the authors and the discussion is appropriate. There are some questions and comments, which I believe should be clarified and/or discussed in the manuscript:
- How was the functional prediction for missense variants performed?
- Why were only exonic GSRs assessed for their functional effect? Intronic rearrangements might affect gene function as well. The number of genomic rearrangements might coincide with defects in the DNA repair genes. Did the authors observe any effects?
- Why was Gleason score not taken into account in the multivariate analysis? I would suggest adding it into the model or mention reasons for omitting.
- Finally, the only independent prognostic genetic factor across both cohorts seems to be TP53. Authors should elaborate on how PCI should be used in addition to TP53 and what the additional benefit of the 39-gene panel might be. In particular as modern ctDNA sequencing methods may encompass >300 genes and can increase the PCI accordingly.
Minor:
- Page 6 Line 169: „PCI relatively tumour mutational burden“. Meaning?
Reviewer 3 Report
In my opinion a very interseting study with sufficient and well presented data. Great effort with good results. No further comments on my side. I enjoyed reading the manuscript.
Reviewer 4 Report
De Laere and colleagues report on the association between an increasing number of ‘pathway level’ genomic aberrations detected in ctDNA and adverse clinical outcomes in mCPRC. The findings are interesting but clarification of some details is required.
- How were the data from the two cohorts amalgamated? Was read level data downloaded and analysed using a common pipeline? How were variants called? What was the minimum necessary evidence required to support a positive call?
- It has been reported that a large number of positive ctDNA aberrations may be false positives (Razavi et al., Nat Med 25:1928-1937, 2019), including in a previous analysis (or the same samples?) by the authors (Mayrhofer et al., Genome Med. 2018 Nov 21;10(1):85). How was this handled in the current analysis?
- It is not clear if increased pathway complexity is a surrogate of total tumour burden - so higher cancer volume leads to greater ctDNA levels which increases the likelihood of detecting aberrations in multiple pathways. The volume of disease (number of metastatic sites, or even low- vs. high-volume disease) should be included in the MV analysis.
- If each cohort is considered alone and includes non-overlapping genomic regions, does the observation of poorer prognosis with increasing pathway complexity hold?
Reviewer 5 Report
The authors have done a comprehensive analysis on metastatic castration resistant prostrate cancer. They show that increased pathway complexity can be used as a prognostic biomarker in mCRPC.
- The introduction needs to be expanded to include more details. The authors can write more about existing genomic profiling in mCRPR and patient heterogeneity observed, more about their prior discoveries and can also include information about driver mutations and relevant signalling pathways (including AR).
- Figure S2 (showing ctDNA landscape) can be included as a main figure.
- The authors have to discuss figure 3 more. To me that is their important data and that is not very well described. Describe the figures more and explain your inferences better.
- Along with driver mutations or PCI, do these patients have additional symptoms or co-morbidities that affect PFS or OS?
Round 2
Reviewer 1 Report
I appreciate the change in the new version.
Reviewer 4 Report
Concerns have been adequately addressed